# *Morus alba* Prevented the Cyclophosphamide Induced Somatic and Germinal Cell Damage in Male Rats by Ameliorating the Antioxidant Enzyme Levels

**DOI:** 10.3390/molecules26051266

**Published:** 2021-02-26

**Authors:** Abhijit Ghosh, Syed Imam Rabbani, Syed Mohammed Basheeruddin Asdaq, Yahya Mohzari, Ahmed Alrashed, Hamdan Najib Alajami, Awad Othman Aljohani, Abdullah Ali Al Mushtawi, Majed Sultan Alenazy, Rakan Fahad Alamer, Abdulmajead Khalid Alanazi

**Affiliations:** 1Department of Pharmacology, Al-Ameen College of Pharmacy, Bangalore 560027, India; jamy47mjn@gmail.com; 2Department of Pharmacology and Toxicology, College of Pharmacy, Qassim University, Buraydah 51452, Saudi Arabia; 3Department of Pharmacy Practice, College of Pharmacy, AlMaarefa University, Dariyah, Riyadh 13713, Saudi Arabia; 4Clinical Pharmacy Department, King Saud Medical City, Riyadh 12746, Saudi Arabia; yali2016@hotmail.com; 5Pharmaceutical Services Administration, Inpatient Department, Main Hospital, KFMC, Riyadh 11564, Saudi Arabia; emadasdaq@gmail.com; 6Pharmaceutical Services Administration, King Saud Medical City, Ministry of Health, Riyadh 12746, Saudi Arabia; ph.hamdan@gmail.com (H.N.A.); A.Aljehani@ksmc.med.sa (A.O.A.); a.almushtawi@ksmc.med.sa (A.A.A.M.); m.alenzy@ksmc.med.sa (M.S.A.); Rakanfalamer@gmail.com (R.F.A.); Ph.me1@hotmail.com (A.K.A.)

**Keywords:** *Morus alba*, micronucleus test, sperm analysis, antioxidant

## Abstract

Cytogenetic analysis is essential to determine the effect of mutagens and antimutagens on genetic material. This study was done to evaluate the protective effect of root bark extract of *Morus alba* (*M. alba*) against cyclophosphamide induced somatic and germinal cell damage in male rats. The ethanolic extract of *M. alba* (0.25, 0.5 and 1 g/kg, 2 weeks) was evaluated against cyclophosphamide (75 mg/kg, single dose) induced nuclear damage. The sampling was done after 48 h of the clastogen treatment. The somatic and germinal nuclear damage was studied by bone marrow micronucleus and sperm analysis, respectively. Serum superoxide and catalase levels were estimated to determine the antioxidant status in each group. The results were analyzed statistically to find the significant variation. The administration of *M. alba* for 2 weeks suppressed dose-dependently the changes induced by cyclophosphamide. *M. alba* (0.5 g/kg) decreased the frequency of micronucleated erythrocyte, sperm shape abnormality and enhanced the sperm count, sperm motility and polychromatic-normochromatic erythrocytes ratio significantly (*p* < 0.05) in comparison with the cyclophosphamide treated group. The highest tested dose of *M. alba* (1 g/kg) produced more prominent suppression (*p* < 0.01) in the cyclophosphamide-induced somatic and germinal cell defects. The results also showed significant (*p* < 0.05) improvement in the serum antioxidant enzymes levels with *M. alba* when compared with the challenge group. The lower dose of *M. alba* extract (0.25 g/kg) prevented the CP-induced changes but was found to be statistically insignificant. Therefore, antimutagenic potential of the high dose of the extract of *M. alba* is possibly due to its antioxidant nature. The ability of the *M. alba* extract to prevent the nuclear damage could play an important role in overcoming several mutational defects that are associated with anticancer chemotherapy.

## 1. Introduction

Damage to DNA by anticancer chemotherapeutic agents is likely to be the major cause of occurrences of secondary tumors [1]. Cytogenetic analysis is one of the most valuable methods for studying the effect of mutagens and antimutagens on the genetic material. In-vivo micronucleus test of the bone marrow in experimental animal is one of the vital techniques to evaluate somatic cell mutations in mammals [2]. The quantitative and qualitative analysis of sperms is considered a rapid screening method for agents that can produce defects in the male germinal cells [3]. Together, these two tests are reported to be the established methods to assess the mutagenic potential of a compound and preventions on somatic and germinal cells [2,3].

Cyclophosphamide is a nitrogen mustard alkylating agent commonly used in the treatment of several types of cancer. Cyclophosphamide exhibits its mechanism by cross-linking the strands of DNA and RNA and by inhibiting protein synthesis [4]. The ability of cyclophosphamide to interfere in the structure and function of nucleic acid is responsible for the cytotoxic and cytogenetic actions. The nuclear damage has the tendency to cause mutation, which in-turn can contribute to several complications including heart ailments, neurological disorders, and cancer [5]. One of the recent approaches in anticancer chemotherapy is to add a known antioxidant/antimutagens so as to minimize the long-term complications without compromising the potency of the anticancer effect [6]. 

*Morus alba* (Family: *Moraceae*) commonly called Mulberry is widely distributed in Asian countries. The extract of *Morus alba* (*M. alba*) is reported for its hypoglycemic, antihypertensive and bacteriostatic properties [7]. The herb is used as diuretic, anticancer and expectorant agent [8]. Published literature also demonstrates the free radical scavenging action and antioxidant effects of leaves and root barks of *M. alba* extracts [9]. The studies have also reported that the extract of the plant possess antihyperlipidemic, antiasthmatic, cardioprotective, and cognitive enhancement activities [10]. Phytochemical studies revealed that flavonoids, cumarins, phenols, and terpenols are the main bioactive constituents in *M. alba* [11]. Some of the components that were isolated and characterized from the extract of *Morus alba* are biphenyl-furocoumarin (phenolic) and morflavanone (flavanoid). These compounds have been reported to possess cardioprotective properties [12]. Previous studies suggested the presence of flavonoids and phenolic acid in the *M. alba* extract for their antioxidant, chemopreventive, antimutagenic, and anticarcinogenic effects [13]. The safety profile of the ethanolic extract of *Morus alba* is established in rodents, where the 14-days treatment did not produce significant cytogenetic damage [14]

Medicines derived from natural sources per se are devoid of major toxic manifestations and their addition is reported to reduce the complications of known anticancer drugs such as cyclophosphamide. Such combinations involving the plant derived products were found to improve the treatment compliance and enhance the prognosis of the therapy [15]. Since the antimutagenic potential of *Morus alba* is not well documented in the literature, this research was planned to evaluate the protective effect of 70% ethanolic extract of *Morusalba* root barks against cyclophosphamide-induced somatic and germinal cell nuclear damage in Wistar rats. 

## 2. Results

### 2.1. Effect of the Pretreatment of M. alba Extract on the Frequency of Micronucleated Erythrocytes in Cyclophosphamide Treated Animals 

Administration of cyclophosphamide to the Wistar rats produced a significant (*p* < 0.001) increase in the polychromatic erythrocyte (PCEs) and normocromatic erythrocytes (NCEs) and decreased the P/N (PCEs/NCEs) ratio (Table 1). The pretreatment of *M. alba* at 0.25g/kg did not alter significantly the frequency of micronucleated erythrocytes and P/N ratio after the administration of cyclophosphamide. Administration of *M. alba* at 0.5 g/kg to the cyclophosphamide treated animals causes a significant (*p* < 0.05) decrease in the population of micronuclei in both PCEs and NCEs and increases the P/N ratio in comparison with the positive control group. Further, when *M. alba* was tested at 1 g/kg, the inhibition (PCEs, NCEs) and increase in the P/N ratio against the cyclophosphamide-induced nuclear damage was found to be more prominent (*p* < 0.01) when comparison was done with cyclophosphamide-treated animals. α-tocopherol tested as a standard antioxidant exhibited a significant (*p* < 0.001) reduction in the frequency of micronucleated PCEs and NCEs. The treatment also reversed (*p* < 0.001) the changes induced by cyclophosphamide on P/N ratio. However, *M. alba* tested alone at 1 g/kg produced no change on the frequency of micronucleated erythrocytes and P/N ratio when compared with control animals (Table 1). 

### 2.2. Effect of Pretreatment of M. alba Extract on Sperm Morphology and Sperm in Cyclophosphamide Treated Animals

Cyclophosphamide (75 mg/kg) tested as a clastogenic agent produced significant (*p* < 0.001) reduction in the sperm count (Appendix A) and enhanced the sperm shape abnormality (Appendix A) in comparison with the control group. The low dose of *M. alba* (0.25 g/kg) did not alter the changes induced by cyclophosphamide in the sperm abnormality. However, administration of *M. alba* at 0.5 g/kg caused significant (*p* < 0.05) reduction in the cyclophosphamide mediated changes on the sperm count and sperm shape abnormalities. Treatment of *M. alba* at 1 g/kg produced further improvement (*p* < 0.01) in the protective effect against the germinal cell toxicity induced by cyclophosphamide. The administration of α-tocopherol, as a standard antioxidant produced significant (*p* < 0.001) reduction in the sperm abnormality induced by cyclophosphamide. Further, administration of *M. alba* (1 g/kg) to the normal animals did not produce any significant change on these parameters (Table 2).

### 2.3. Effect of Pretreatment of M. alba Extract on Sperm Motility in Cyclophosphamide Treated Animals 

The results indicated that cyclophosphamide treatment reduced significantly (*p* < 0.01) the sperm motility (Appendix A) in all the tested time intervals compared to normal group. *M. alba* at the low dose (0.25 g/kg) did not show significant improvement in the number of motile cells, however at medium and high doses (0.5 and 1 g/kg, respectively) *M. alba* improved the sperm mobility at different intervals. The medium dose of *M. alba* significantly (*p* < 0.05) increased the number of viable cells whereas high dose of *M. alba* (1 g/kg) produced further enhancement (*p* < 0.01) in this effect when compared to cyclophosphamide treated controls. Additionally, α-tocopherol reduced (*p* < 0.001) the cyclophosphamide mediated defects by enhancing the number of viable spermatozoa cells. The highest tested dose of *M. alba* (1 g/kg) when administered to normal animals did not significantly change the sperm motility when compared with the control group (Figure 1).

### 2.4. Effect of Pretreatment of M. alba Extract on Serum SOD Levels in Cyclophosphamide Treated Animals 

*M. alba* at 1 gm did not produce significant change, however, cyclophosphamide treatment decreased (*p* < 0.001) the serum level of SOD compared to the control. At the low dose, *M. alba* (0.25 g/kg) caused no alteration in the SOD level in the cyclophosphamide administered group, however, when the dose of *M. alba* was increased, a concentration dependent enhancement in serum SOD level was observed. The high dose of *M. alba* (1 g/kg) showed better antioxidant action (*p* < 0.01) than the lower tested dose of *M. alba* (0.5 g/kg, *p* < 0.05) in comparison to cyclophosphamide control. Administration of α-tocopherol to the cyclophosphamide challenged animals indicates that the treatment significantly (*p* < 0.001) improved the antioxidant status (Figure 2).

### 2.5. Effect of Pretreatment of M. alba Extract on Serum Catalase Levels in Cyclophosphamide Treated Animals 

The results obtained for the estimation of serum catalase levels are presented in Figure 3. The administration of cyclophosphamide to the experimental rats significantly (*p* < 0.001) suppressed the serum catalase level in comparison to the control. Prior treatment of low dose of *M. alba* (0.25 g/kg) produced no significant variation in the serum catalase levels. The extract of *M. alba* at moderate dose (0.5 g/kg) significantly (*p* < 0.05) enhanced the catalase concentration and high dose of *M. alba* (1 g/kg) demonstrated more improvement (*p* < 0.001) in the serum antioxidant status. In addition to this, α-tocopherol pretreatment significantly (*p* < 0.001) elevated the catalase level in the serum in comparison to the cyclophosphamide-alone animals. *M. alba* (1 g/kg) when tested alone did not alter significantly the serum catalase levels in normal animals (Figure 3).

## 3. Discussion

Cyclophosphamide is a known mutagen and produces the cytotoxic effect by an electrophilic attack on the nucleophilic site in the DNA [5]. Administration of cyclophosphamide to the normal animals in this study increased the micronucleated erythrocytes and reduced the P/N ratio. The cyclophosphamide treatment also decreased the total sperm count and sperm motility and enhanced the defects in the sperm morphology.

The fractions of main nucleus that remain unaffected after the nuclear damage and lie in the cytoplasm is micronuclei. Increase in the frequency of micronuclei indicates the extent of nuclear defects the test substance has produced in the erythrocytes. The P/N ratio in normal condition is found to be 1 and suppression in its value suggests the cytotoxic effect of the compound [16,17]. The reduction in the total sperm count and increase in sperm shape abnormalities is reported to indicate defective spermatogenesis and structural damage to spermatozoa, respectively, and both are indicative of the nuclear abnormalities the germinal cell has undergone [5,18]. The diminished sperm motility has been linked to the ability of the compound to affect the viability of the sperm cells [19]. The micronucleus and sperm abnormality assays are reported to be the common techniques employed to detect somatic and germinal cell mutations [2,17].

The finding from the present study confirms the earlier reports that cyclophosphamide produces both somatic and germinal cell damage in mammalian system [6]. Further, the decreased levels of two important antioxidant enzyme viz., SOD and CAT (Figure 2 and Figure 3) suggests that the cytogenetic damage produced by cyclophosphamide most likely involves the generation of reactive oxygen species (ROS) and the mechanism suggested is the activation of microsomal enzymes by cyclophosphamide and its metabolic acrolein [6,9].

Oxidative stress is reported to cause both DNA adduct as well as the reproductive damage. The modification in the nuclear part of the somatic cells contributes in carcinogenesis, neurological defects, aging, etc., while the germinal cell damage results in alteration of genes leading to neonatal defects including childhood cancer [1,5]. Hence analyzing the nuclear defects and germinal cell damage and their prevention assumed prime importance in minimizing the disorders related to exogenous mutagens [7,8].

In this study, administration of α-tocopherol minimized the cyclophosphamide related changes on the frequency of micronuclei, sperm shape abnormality, sperm count, sperm motility, besides elevating the antioxidant enzyme levels (Table 1 and Table 2 and Figure 1, Figure 2 and Figure 3). α-tocopherol being an antioxidant, in the earlier studies had reduced the oxidative damage in both somatic cells and germinal cells [20] and this can be evidence from the present study where α-tocopherol elevated SOD and CAT and minimized the DNA damage and male reproductive cell abnormalities (Table 2, Figure 1, Figure 2 and Figure 3). As reported, SOD is an enzyme that catalyzes the dismutation of superoxide ion into oxygen and H_2_O_2_, while CAT promotes efficient conversion of H_2_O_2_ to water and molecular oxygen, collectively protecting cells from the toxic effects of oxidants [6,19,21]. These studies suggest that compounds possessing the antioxidant property can provide beneficial effects in reducing the cyclophosphamide mediated cytogenetic damage in the host system [6,20].

Another important finding of this study is that administration of *M. alba* (0.5 and 1 g/kg) reduced the cyclophosphamide-mediated nuclear injury, sperm abnormalities apart from enhancing the antioxidant status (Table 1 and Table 2 and Figure 1, Figure 2 and Figure 3). In earlier studies, it was estimated that mulberry plants (*M. alba*) contain a high content of anthocyanine pigment in addition to sugar and acid [12]. One of the characteristics of anthocyanines is the changes in solution coloration in response to the pH of the environment. Anthocyanines reported to exhibit a potent inhibitory effect on the generation and release of free radicals by human granulocytes in vitro, as measured by nitroblue tetrazolium (NBT) reduction test. Anthocyanins showed an antimutagenic influence through multiple mechanisms, one of which could be a limitation of free radicals’ involvement in mutagenesis [22]. The known antioxidant property of anthocyanins has been related to its structure, namely the oxonium ion in the C ring. The findings have suggested that the antioxidant functions of anthocyanins are related to the aglycone moiety, and have also been linked to cyanidin and glycosides. However, the number of sugar residues at the 3-position, the oxidation state of the C ring, the hydroxylation and methylation pattern, as well the acylation by phenolic acids are reported to be important determinants for the expression of antioxidant effects [23].

Based on this information, it can be suggested that *M. alba* prevented the cyclophosphamide-induced somatic and germinal cells damage by scavenging the free radical generated by cyclophosphamide. Further studies in this direction might explore the precise mechanism and the antimutagenic potential of *M. alba* in limiting the disorders related to cancer chemotherapy.

## 4. Materials and Methods

### 4.1. Chemicals:

α-tocopherol was purchased from Titan Biotech LTD, Mumbai, India and cyclophosphamide from Sigma Aldrich Ltd, Mumbai, Maharashtra, India. Other chemicals and stains used in this study were purchased from local supplier and were of analytical grade.

### 4.2. Animals

Laboratory bred male Albino Wistar rats (around eight weeks old, 140 ± 10 g) maintained under standard laboratory with water and ad libitum were used in this study. Prior approval from the Institutional Animal Ethics Committee (AACP/IAEC/P-18/2018) was obtained. The animals experienced 2 weeks of acclimatization in the lab conditions before starting the experiment. All the studies were done in a humane manner. Diseased and unhealthy rats were returned to the central animal house for further care.

### 4.3. Extraction of Active Constituents from Morus alba

Dried *Morus alba* root barks were collected and authenticated in Indian Institute of Horticulture Research, Bangalore (Ref No-F.No.E and T/ATIC/Misc/2018-19/75). The dried root bark was powdered and passed through sieve no. 10. The drug was extracted with the solvent consisting of ethanol and water in the ratio of 7:3 [24]. The extract was collected, filtered, and concentrated using a rotary vacuum evaporator. Dried extract was weighed and suspended in vehicle (1% *w/v* CMC) and administered to animals according to the dose and body weight.

### 4.4. Dose and Treatment

The animals were divided mainly into three groups: control, challenge and treatment. The control and *M. alba* (1 g/kg) groups comprised six animals, while cyclophosphamide treated group included eight animals. A mortality rate of 10% and 5% was observed in cyclophosphamide and cyclophosphamide + *M. alba* (0.25 g/kg) groups, respectively. The control group received saline (0.5 mL/kg) while the challenge group was treated with cyclophosphamide (75 mg/kg, i.p.) [25]. In the treatment group, *M. alba* was tested in three doses (0.25, 0.5 and 1 g/kg, p.o). *M. alba* was administered daily for 2 weeks and on the 14th day, cyclophosphamide was administered. A solvent control group was also performed using 1% *w/v* CMC, administered for 2-weeks orally. α-tocopherol (50 mg/kg, p.o.) was used as an internal standard antioxidant agent [16].

### 4.5. Bone Marrow Micronucleus Test

Bone marrow micronucleus test was conducted according the modified method of Schimid [17]. At the end of treatment duration, animals were sacrificed by cervical dislocation under light ether-anesthesia. The femur and tibia bones were exposed, and the bone marrow suspension was prepared in 5% *w/v* bovine serum albumin (BSA). The extract fluid was centrifuged at 1000 rpm for 8 min and the pellet was resuspended in a definite quantity of BSA. A cleaned and dried microscopic glass was collected, and a drop of the suspension was placed it to prepare a smear. The single layered smear was fixed in absolute methanol, stained with May–Grunwald–Giemsa and micronuclei were identified in polychromatic erythrocyte (PCEs) and normochromatic erythrocytes (NCEs) [26]. The presence of micronuclei was counted using oil-immersion objective in about 2000 PCEs and corresponding number was recorded in NCEs.

### 4.6. Sperm Analysis

#### 4.6.1. Sperm Shape Abnormality Assay

The sperm shape abnormality was done as per the procedure described by Shruthi and Vijayalaxmi (2016) [18]. The dissected cauda epididymis was minced in phosphate buffer (pH 7.2) and stained for half an hour with 1% eosin. A smear was prepared on the clean glass slide and about 1000 spermatozoa cells per animal were screened under microscope (40×) (Micron instrument industries, Ambala, Haryana, India) to identify the abnormality. Six types of sperm shape abnormalities such as curved, banana-shaped, headless, amorphous, double-tailed, and double-headed were identified from the screened spermatozoa [26].

#### 4.6.2. Total Sperm Count

The procedure reported by D’Souza (2004) was used for total sperm count [27]. The cauda epididymis collected in 1.0 mL of phosphate buffer and spermatozoa were released from the cut epididymis by gentle pressure using fingers. Muslin cloth was used for filtering the suspension. One to two drops of 1% aqueous eosin yellow stain was added to the filtrate. The sperm cells were counted using the four WBC chambers of Neubauers’ slide.

#### 4.6.3. Sperm Motility

The skin of the scrotum was cut to open tunica vaginalis so that epididymis was removed and flushed in 1 mL of warm buffer saline (pH 7.4) at 37 °C to make suspension of spermatozoa. A suspension drop was placed on the slide and observed under microscope (10×). The motility of sperm was assessed at different time intervals [28].

### 4.7. Estimation of Antioxidant Enzymes

#### 4.7.1. Superoxide Dismutase (SOD)

The principle for measuring the SOD depends on the detecting the O_2_^-^ generated during auto-oxidation of hydroxylamine. During the oxidation, nitro blue tetrazolium (NBT) can be detected in the presence of EDTA calorimetrically at 560 nm. The concentration of SOD is expressed as mg protein/mL [21].

#### 4.7.2. Catalase

The catalase activity was obtained by estimating the amount of decomposition of H_2_O_2_ at 240 nm in an assay mixture containing the phosphate buffer (0.25 M, pH 7). The catalases of the decomposition of 1 mM H_2_O_2_ per min at 37 °C was calculated as one international unit of catalase and is expressed as mg protein/mL [19].

### 4.8. Statistics

The analysis of the results was done using one-way ANOVA test followed by post-hoc analysis by Bonferroni test. Data obtained from the experiments was analyzed and compared between groups. A minimum of six replicates were performed for each analysis. *p* value <0.05 was used to determine the significance of the results. All the values represented in the study are expressed as Mean ± SD.

## 5. Conclusions

The present study indicates that the ethanolic extract of *Morusalba (M. alba)* root bark possess antimutagenic effect against the cyto-nuclear damage caused by cyclophosphamide (cyclophosphamide). The extract prevented the incidences of micronuclei formation, sperm shape abnormality and enhanced the diminished sperm count and sperm motility and erythropoiesis induced by the clastogen. The antioxidant enzyme estimation in serum indicated that *M. alba* has the potential to enhance the level of SOD and catalase. The ability of the *M. alba* extract to prevent the nuclear damage caused by the known mutagen could play an important role in overcoming several mutational defects that are associated with anticancer chemotherapy.

## Figures and Tables

**Figure 1 molecules-26-01266-f001:**
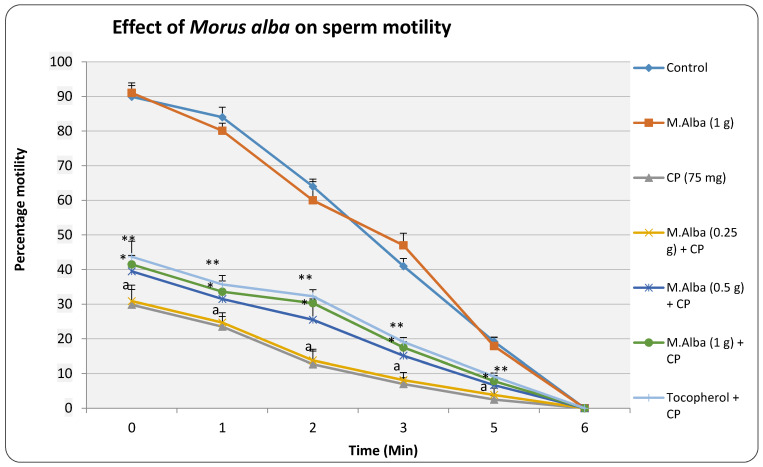
Effect of pretreatment of *M. alba* extract on sperm motility in cyclophosphamide treated animals. (P.S: *M. alba—Morus alba*, CP—cyclophosphamide, α-toco—α-tocopherol); values are represented as Mean ± SD, N = 6. Statistics: one-way ANOVA followed by Bonferroni; a *p* < 0.001 compared with control, * *p* < 0.05, ** *p* < 0.01 compared with cyclophosphamide group.

**Figure 2 molecules-26-01266-f002:**
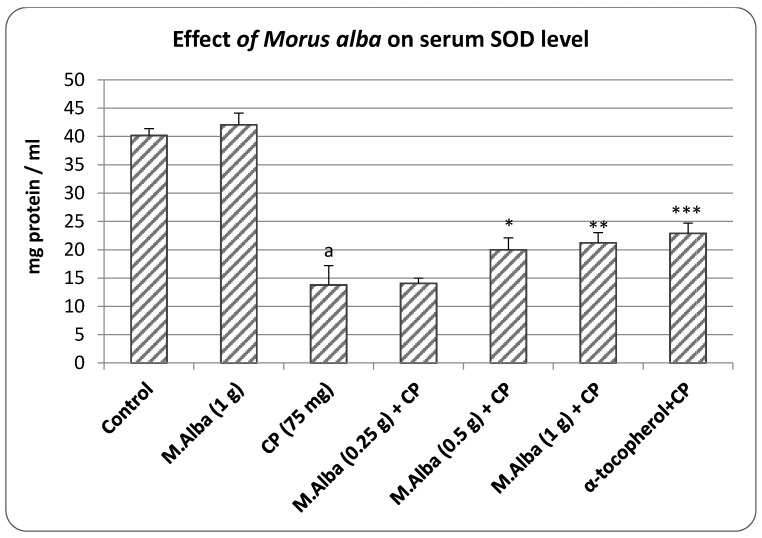
Effect of pretreatment of *M. alba* extract on serum SOD levels in cyclophosphamide treated animals. (P.S: *M. alba*—*Morus alba*, CP—cyclophosphamide, α-toco—α-tocopherol); values are represented as Mean ± SD, N = 6. Statistics: one-way ANOVA followed by Bonferroni; a *p* < 0.001 compared with control, * *p* < 0.05, ** *p* < 0.01, *** *p* < 0.001 compared with cyclophosphamide group.

**Figure 3 molecules-26-01266-f003:**
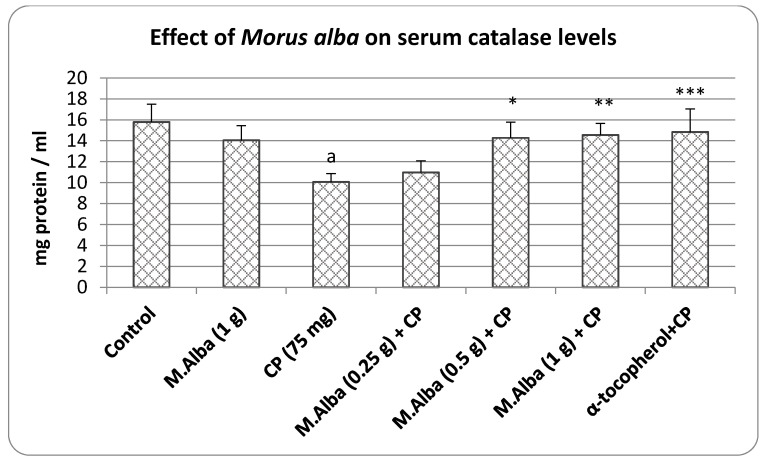
Effect of pretreatment of *M.*
*alba* extract on serum catalase levels in cyclophosphamide. Statistics: one-way ANOVA followed by Bonferroni; a *p* < 0.001 compared with control, * *p* < 0.05, ** *p* < 0.01, *** *p* < 0.001 compared with cyclophosphamide group.

**Table 1 molecules-26-01266-t001:** Effect of the pretreatment of *Morus alba* extract on the frequency of micronucleated erythrocytes in cyclophosphamide treated animals.

Bone Marrow Micronucleus Test	Control	*M. alba* (1 g/kg)	Cyclophosphamide (CP-75 mg)	*M. alba* (0.25 g/kg) + CP	*M. alba* (0.5 g/kg) + CP	*M. alba* (1 g/kg) + CP	α-Toco (50 mg) + CP
Micronuclei in PCE (%)	0.52 ± 0.02	0.55 ± 0.04	1.09 ± 0.07 ^a^	1.03 ± 0.02	1.01 ± 0.02*	1.00 ± 0.02**	0.96 ± 0.02***
Micronuclei in NCE (%)	0.49 ± 0.02	0.50 ± 0.02	1.74 ± 0.19 ^a^	1.58 ± 0.04	1.55 ± 0.02 *	1.49 ± 0.02 **	1.42 ± 0.02 ***
P/N Ratio	1.05 ± 0.04	0.99 ± 0.04	0.62 ± 0.04 ^a^	0.60 ± 0.02	0.55 ± 0.02 *	0.53 ± 0.02 **	0.49 ± 0.02 ***

(P.S: *M. alba*—*Morus alba*, CP—cyclophosphamide, α-toco—α-tocopherol); values are represented as Mean ± SD, N=6. Statistics: one-way ANOVA followed by Bonferroni; ^a^
*p* < 0.001 compared with control, * *p* < 0.05, ** *p* < 0.01, *** *p* < 0.001 compared with cyclophosphamide group.

**Table 2 molecules-26-01266-t002:** Effect of pretreatment of *Morus alba* extract on sperm morphology and sperm in cyclophosphamide treated animals.

Sperm Analysis	Control	*M. alba* (1 g/kg)	Cyclophosphamide (CP-75 mg)	*M. alba* (0.25 g/kg) + CP	*M. alba* (0.5 g/kg) + CP	*M. alba* (1 g/kg) + CP	α-Toco (50 mg) + CP
Sperm count (10^6^)	38.12 ± 1.88	39.03 ± 2.09	22.15 ± 1.27 ^a^	22.17 ± 0.61	25.73 ± 1.10 *	26.95 ± 2.80 **	27.47 ± 2.42 ***
Sperm shape abnormality (%)	1.88 ± 0.19	1.69 ± 0.14	5.03 ± 0.31 ^a^	4.78 ± 0.31	4.23 ± 0.71 *	3.89 ± 0.31 **	3.82 ± 0.08 ***

(P.S: *M. alba*—*Morus alba*, CP—cyclophosphamide, α-toco—α-tocopherol); values are represented as Mean ± SD, N = 6. Statistics: one-way ANOVA followed by Bonferroni; ^a^
*p* < 0.001 compared with control, * *p* < 0.05, ** *p* < 0.01, *** *p* < 0.001 compared with cyclophosphamide group.

## Data Availability

Data is contained within the article.

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
