# Peer review of "Morus alba* Prevented the Cyclophosphamide Induced Somatic and Germinal Cell Damage in Male Rats by Ameliorating the Antioxidant Enzyme Levels"

_molecules, 2021, doi:10.3390/molecules26051266_

Round 1
Reviewer 1 Report
Manuscript “Preventive effect of Morus alba extract on cyclophosphamide-in-2 duced somatic and germinal nuclear damages in male rats” presented by Abhijit Ghosh et al. describes results of investigation of influences of plant Morus alba extract on systemic detrimental effects of anticancer agent cyclophosphamide. Several biological parameters including enzyme levels, spermatozoa motility and morphology, and somatic and germinal nuclear damages are investigated. Described outcome of investigation is an important contribution in finding additional therapeutics in cancer treatments. The manuscript is well written, composed, and illustrated. However, there are several shortfalls, which should be addressed. The main challenge is the lack of emphasis on results that demonstrate the highest efficiency of treatment e.g. recovery of catalase activity (by the way use the term “levels” for enzyme activity is misleading due to its similarity to expression levels). Some of the observed effects, despite their statistical significance, look minuscule compared to a total number of events. A better explanation of why such demonstration is important should be presented. Additional challenge relates to a clear description of t number of technical replicates and events (cells) measured. How representative is collected data if only 1000 spermatozoa analyzed from the millions present?
Additional comments
Title
- The title is mentioning only nuclear damages omitting most of effects described in the manuscript including enzyme levels and spermatozoa morphology and motility. A more general title should be presented.
Abstract
- Cytogenetic analysis is mentioned as a prime approach. However, what results can be attributed to this approach e.g. decreased the frequency of micronucleated erythrocyte, sperm shape abnormality, and enhanced the sperm count, sperm motility, and polychromatic-normochromatic erythrocytes ratio?
- What is “the antioxidant status”? Is the determination of levels (activities) of two enzymes sufficient for establishing such status? What about glutathione and vitamin E?
- What is “significant (p<0.05) improvement in the serum antioxidant enzymes”? Normalization of levels and/or activities”?
- It is worth mentioning which doses did not produce significant results.
Introduction
- An additional paragraph should be added which would describe currently used pharmaceuticals that have and have no targeted in this work Cyclophosphamide. This will clarify why this drug or dug type is important to study.
Results
- Figure S1 shows just a single image and cannot be used for evaluation of described results requiring statistical analysis “significant 77 (p<0.001) increase in the polychromatic erythrocyte (PCEs) and normocromatic erythro-78 cytes (NCEs) and decreased the P/N (PCEs/NCEs) ratio (Figure S1)”. Table 1 should be listed at this position.
- Description of results in text somewhat misleading. Percentages of changes are calculated from percentages of differences. For example, the difference of 0.09% in the table corresponds to a 7.3% description of change in text. Are data significant statistically? Yes. Is it meaningful clinically or biologically? Very questionable. Presentation of data in numbers of observed cells with modifications, probably, is more convincing.
- Table 2. Describe “Sperm shape abnormality” dimensionality. Is it 1.88 per 38.12 million?
- Fig 1. One trace has no label in legend.
Materials and Methods
- Please clarify, I male animals were used why pregnant rats were returned?
- The animal section may also contain information on the number of animals used in each experimental set. Please describe any outliers and animals that did not survive during treatments. Such information can be described in section “2.4. Dose and treatment:” as well.
- How many micronuclei were examined per data set?
- 1000 spermatozoa were examined per what – animal or measurement? How many measurements were performed per animal for each condition? Table 2 lists Sperm shape abnormalities. Does it mean that 1.88 relates to 1.88 per 1000?
- Describe how many technical replicates were performed for each measurement.
Author Response
All the comments were addressed and the summary is represented in the following table:
|
Reviewers’ Comment |
Reply to comments |
Page number |
|
Title: The title is mentioning only nuclear damages omitting most of the effects described in the manuscript including enzyme levels and spermatozoa morphology and motility. A more general title should be presented. |
Morus alba prevented the cyclophosphamide induced somatic and germinal cell damages in male rats by ameliorating the antioxidant enzyme levels |
Page # 1 |
|
Abstract: Cytogenetic analysis is mentioned as a prime approach. However, what results can be attributed to this approach e.g. decreased the frequency of micronucleated erythrocyte, sperm shape abnormality, and enhanced the sperm count, sperm motility, and polychromatic-normochromatic erythrocytes ratio? |
Since micronucleus test and sperm shape abnormality are the established methods to determine the cytogenetic analysis, this term was used as a prime approach in the study. |
Page # 1 |
|
Abstract: What is “the antioxidant status”? Is the determination of levels (activities) of two enzymes sufficient for establishing such status? What about glutathione and vitamin E? |
It is agreed that the antioxidant status can be established by testing the level of various enzymes such as SOD, catalase, GSH, LPO, vitamin E, etc, but some previous studies have reported the effect with two enzyme estimations also (Reference: Sharma I, Aaradhya M, Kodikonda M, Naik PR. Antihyperglycemic, antihyperlipidemic and antioxidant activity of phenolic rich extract of Brassica oleraceaevargongylodes on streptozotocin induced Wistar rats. Springerplus. 2015 May 3;4:212). Moreover, the main objective of the present study is to evaluate the anti-mutagenic potential of the M. alba extract, hence, as a supportive mechanism, we measured the levels of two antioxidant enzymes. |
Page # 1 |
|
Abstract: What is “significant (p<0.05) improvement in the serum antioxidant enzymes”? Normalization of levels and/or activities”? |
The statement refers to the level of antioxidant enzymes in serum. |
Page # 1 |
|
Abstract: It is worth mentioning which doses did not produce significant results. |
The suggestion has been incorporated in the abstract where the lower dose of M. alba extract (0.25 gm/kg) though prevented the CP-induced changes but were found to statistically insignificant. |
Page # 1 |
|
Introduction: An additional paragraph should be added which would describe currently used pharmaceuticals that have and have no targeted in this work Cyclophosphamide. This will clarify why this drug or dug type is important to study. |
A new paragraph is added now to describe the need for testing the drugs of plant origin as they are expected to have fewer adverse effects. |
Page # 2 |
|
Results: Figure S1 shows just a single image and cannot be used for evaluation of described results requiring statistical analysis “significant 77 (p<0.001) increase in the polychromatic erythrocyte (PCEs) and normocromatic erythro-78 cytes (NCEs) and decreased the P/N (PCEs/NCEs) ratio (Figure S1)”. Table 1 should be listed at this position. |
Improper positioning of the figure-1 is removed in the revised manuscript. Table 1 is mentioned instead. |
Page # 2 |
|
Results: Description of results in text somewhat misleading. Percentages of changes are calculated from percentages of differences. For example, the difference of 0.09% in the table corresponds to a 7.3% description of change in text. Are data significant statistically? Yes. Is it meaningful clinically or biologically? Very questionable. Presentation of data in numbers of observed cells with modifications, probably, is more convincing. |
For better understanding, the percentage difference values were removed. |
Page # 3 |
|
Results: Table 2. Describe “Sperm shape abnormality” dimensionality. Is it 1.88 per 38.12 million? |
The values are represented in percentage and the same has been incorporated in table 2. |
Page # 4 |
|
Results: Fig 1. One trace has no label in legend |
One label was hidden. Now it is visible in Fig 1. |
Page # 4 |
|
Materials and methods: Please clarify I male animals were used why pregnant rats were returned? |
There was a mistake and the same has been corrected. |
Page # 8 |
|
Materials and methods: The animal section may also contain information on the number of animals used in each experimental set. Please describe any outliers and animals that did not survive during treatments. Such information can be described in section “2.4. Dose and treatment:” as well. |
The details about the number of animals and the mortality rate during experimentation are provided under section 2.4. |
Page # 9 |
|
Materials and methods: How many micronuclei were examined per data set? |
As per the guidelines, 2000 polychromatic erythrocytes (PCEs) and corresponding normochromatic erythrocytes (NCEs) were scanned for the presence of micronuclei. In the control group, around 10 micronucleated PCEs were identified and this number increased to 20 in cyclophosphamide treated rats. Accordingly, the percentage of MN was determined in different groups of treatments. This has been described in the manuscript. |
Page # 9 |
|
Materials and methods: 1000 spermatozoa were examined per what – animal or measurement? How many measurements were performed per animal for each condition? Table 2 lists Sperm shape abnormalities. Does it mean that 1.88 relates to 1.88 per 1000? |
1000 spermatozoa were examined per animal and among these different types of sperm shape abnormalities were recorded. The sperm shape abnormality is represented as a percentage. This has been included in the table. |
Page # 9 |
|
Materials and methods: Describe how many technical replicates were performed for each measurement. |
Each group comprised of 6 – 8 animals. After respective treatments, all the survived animals were subjected to different types of analysis. A minimum of 6 replicates were performed for each measurement. |
Page # 10 |

Reviewer 2 Report
The abstract clearly shows the main aim of the research, the research questions, and the main results and conclusions of the research conducted. The title of the paper is also clear and concise. All the tables and figures are self-descriptive and clearly show the research findings. Moreover, all presented tables and figures are necessary to highlight the main results.
In the discussion, the authors present the research question and give clear explanations of the results obtained, supported by relevant scientific literature.
The materials and methods of the study are clearly described, the protocols of the methods are detailed, and based on the above data, the experiment can be repeated. The conclusion briefly and concisely summarizes the results obtained and is in line with the stated aim of the study.
The literature data cited are current and necessary to compare the results. In the end, the research is original, well presented, scientifically sound, and meaningful, and I conclude that it will be of great interest to the scientific public and will provide a good foundation for future research on this topic.
Based on all of this, I recommend that the Editorial office publish the article without significant changes.
Author Response
Thanks for your comments respected reviewer

Reviewer 3 Report
The present study indicates that the ethanolic extract of Morus alba root bark possesses an anti-mutagenic effect against cytonuclear damage and could be used in cancer treatment. Overall, the paper is well written and could be published. There are some minor comments:
The introduction could be added more information about the usage of Morus alba in health therapy.
The ethanolic extract should be characterized (at least total phenolic contents, antioxidant activity et al).
The toxicity of the extract should be considered.
Author Response
The following table summarized the reply to reviewers’ comment;
|
Reviewers’ Comment |
Reply to comments |
Page number |
|
The introduction could be added more information about the usage of Morusalba in health therapy. |
A new paragraph was added to give more description about the health benefits of Morus alba. |
Page # 2 |
|
The ethanolic extract should be characterized (at least total phenolic contents, antioxidant activity et al). |
Characterization of two isolated components of the ethanolic extract of M. alba and the reported pharmacology activity is provided in the introduction section of the revised manuscript. |
Page # 2 |
|
The toxicity of the extract should be considered. |
The safety profile of the Morus alba is provided in the introduction section of the manuscript |
Page # 2 |
